# COVID-19 Health Crisis Workloads and Screening for Psychological Impact in Nursing Home Staff: A Qualitative and Quantitative Survey

**DOI:** 10.3390/ijerph19074061

**Published:** 2022-03-29

**Authors:** Nesrine Tebbeb, Fanny Villemagne, Thomas Prieur, Solène Dorier, Emmanuel Fort, Thomas Célarier, Luc Fontana, Nathalie Barth, Carole Pélissier

**Affiliations:** 1Occupational Health Service University Hospital Center of Saint-Etienne, 42005 Saint-Etienne, France; nesrinetebbeb@hotmail.fr (N.T.); fanny611@hotmail.fr (F.V.); luc.fontana@univ-st-etienne.fr (L.F.); 2Gérontopôle AURA, 42100 Saint-Etienne, France; thomas.prieur@gerontopole-aura.fr (T.P.); solene.dorier@gerontopole-aura.fr (S.D.); nathalie.barth@gerontopole-aura.fr (N.B.); 3UMRESTTE, Université Lyon 1, Université Gustave Eiffel—IFSTTAR, UMR T 9405, CEDEX 08, 69373 Lyon, France; emmanuel.fort@univ-lyon1.fr; 4Gérontologie Clinique Centre, Hospitalier Universitaire de Saint-Etienne, CEDEX 2, 42055 Saint-Etienne, France; thomas.celarier@chu-st-etienne.fr; 5Chaire Santé des Ainés Ingénierie de la Prévention, Université Jean Monnet de Saint-Etienne, 42100 Saint-Étienne, France; 6Hospital University Center of Saint-Etienne, Université Lyon 1, Université de St Etienne, Université Gustave Eiffel-IFSTTAR, 42005 Saint-Etienne, France

**Keywords:** nursing home staff, COVID 19 health crisis, mental disorders, screening, prevention, quantitative and qualitative study

## Abstract

**Background:** Nursing homes were particularly affected by the COVID-19 pandemic. The purpose of this study was to evaluate qualitatively and quantitatively with the use of a self-reported questionnaire as a tool for screening for mental disorders in nursing home staff. **Methods:** A multicenter epidemiological study was conducted in 12 nursing homes in France with 1117 nursing home staff eligible. Socio-demographic, occupational, and medical data were collected by anonymous self-reported questionnaire using validated scales to assess anxiety/depressive symptoms (HAD scale) and post-traumatic stress disorder (PCL-5). A total of 12 semi-structured interviews were conducted to assess acceptance and expectations for the use of the questionnaire. **Results:** The participation rate was 34.5%. Data from 373 questionnaires were included in the analysis. The questionnaire was well accepted by the participants and met their wishes for prevention action. The sample was 82% female. More than half reported a feeling of powerlessness and lack of time or staffing. The prevalence of anxiety symptoms was 22%, depressive symptoms 10%, and post-traumatic stress 7%. **Conclusions:** This study underlines the interest in screening for mental disorders by self-reported questionnaire and deploying preventive actions in the workplace to reduce stress and facilitate the reconciliation of family and working life in this context of the pandemic.

## 1. Introduction

The year 2020 was marked by the emergence of COVID-19, responsible for a global pandemic. Between 1 March and 29 December 2020, the French Public Health Agency counted 64,068 Covid-19-related deaths in hospitals and facilities for the elderly, compared with an excess of 62,800 deaths from all causes and all places of death over the same period in 2020 compared with 2019 [1]. A total of 10,301 elderly people died in nursing homes between 1 March and 8 June 2020 [2]. Caregivers involved in the care of patients infected with COVID-19 are exposed to stressful situations, intense emotional load linked to the new conditions and organization of work, distress and, sometimes, death of patients and the suffering of families, which can jeopardize their psychological health. Previous studies highlighted the intense psychosocial stress to which nursing home staff are exposed and their links with impaired physical and psychological health [3,4,5,6].

Nursing home staff had to react to the influx of patients infected by COVID-19, presenting severe forms, leading to a dramatic increase in mortality in healthcare services and requiring strong mobilization of caregivers. The emergence of this new virus created a climate of anxiety, concern and uncertainty for healthcare providers and also for the general public [7].

As understanding of viruses and mechanisms of dissemination was poor at the beginning of the pandemic in 2020, the very mention of COVID-19 generated great anxiety. On a global scale, the WHO estimates that 30% to 50% of populations affected by the disaster have suffered from some form of psychological distress [8].

Mental health providers define trauma as a stressful occurrence outside the range of usual human experience that would be markedly distressing to almost anyone [9]. Exposure to traumatic and stressful events can lead to acute stress disorder and finally to post-traumatic stress disorder (PTSD) if symptoms persist [10]. PTSD is a trauma-related disorder that is characterized by the presence of at least one of four symptoms (intrusion, avoidance, negative mood, and cognitive changes), plus arousal and reactivity, for at least 1 month [11].

Numerous studies have demonstrated the presence of stress, anxiety disorder, depression, and post-traumatic stress disorder in many caregivers [8,12,13,14,15].

The French High Authority of Health published recommendations, validated on 7 May 2020, aimed at preventing and identifying occupational suffering in the health sector, with guidance in the context of the COVID-19 crisis [16]. Health professionals are in the first line in the management of the epidemic in healthcare facilities. Considering that humanity is undergoing the most severe pandemic since the Spanish flu, the current COVID-19 pandemic is very likely to promote PTSD, which commonly occurs during major disasters [17,18]. The French National Academy of Medicine, on 8 June 2020, recommended that special long-term attention be paid to the mental health of caregivers involved in the management of COVID-19, who should have a systematic medical examination and 3 years’ follow-up by preventive medicine doctors to identify “possible psychological symptoms that appeared after the acute phase of the health crisis” [15].

Currently, the medical follow-up of caregivers by the occupational health service is based on medical consultations [19]. Due to a lack of occupational physicians in healthcare institutions, systematic consultation would be very difficult to implement in the short term [20]. In this context, it is necessary to facilitate the identification and adapted psychological management of anxiety, depressive, and post-traumatic stress disorder in all staff (nursing and non-nursing) exposed during this pandemic.

The purpose of this study was to evaluate qualitatively and quantitatively the use of a self-reported questionnaire as a tool for screening for mental disorders in nursing home staff.

Study objectives were:to evaluate the acceptance and expectations regarding screening by self-reported questionnaire;to determine the prevalence of anxiety, depressive, and post-traumatic stress disorder in nursing home staff involved in the COVID-19 health crisis;to identify occupational and medical factors associated with anxiety, depression, and post-traumatic stress disorder symptoms in order to adapt prevention actions.

## 2. Methods

### 2.1. Target Population

The target population was nursing home staff involved in the health crisis in the Saint-Etienne area.

Saint-Etienne is a city of 173,089 inhabitants located in the Loire department in France. A total of 72% of the population has a level of education equivalent to or lower than the baccalaureate. In 2019, the median income was 18,410 euros [21].

The directors of 24 nursing homes for the elderly were asked to include their staff, 12 of whom agreed. The participants received clear and comprehensible information on study objectives and procedures and were free to decline participation. A declaration was made to the CNIL data protection commission before starting the study.
Inclusion criteria:–Member of staff having worked in nursing homes for the elderly involved in the health crisis for at least 12 months.Exclusion criteria:–Age under 18 years.

### 2.2. Measurements

In a first step, 12 semi-structured interviews (4 managers, 4 healthcare staff, 3 non-healthcare staff, 1 occupational physician) were conducted. The sample size was determined according to the empirical saturation principle [22]. The interview grid included questions on the acceptability of the questionnaire (theme 1), the conditions under which it was administered (theme 2), and the participants’ expectations regarding psychological care and preventive actions (theme 3). Eligible patients were provided with clearly understandable information on the study protocol and objectives. The information document specified that individual interviews would be recorded to facilitate transcription and that data would be anonymized.

In a second step, an anonymous self-administered questionnaire was provided online via a specific URL and QR code link for each nursing home and in paper format from 26 March to 31 May 2021. Self-administration time was approximately 15 min.

The questionnaire was developed using scientifically validated evaluation scales for anxiety, depression, and PTSD.

HAD scale for assessing anxiety and depression

The Hospital Anxiety and Depression Scale (HADS) is a self-administered questionnaire developed by Snaith and Zigmond in 1983 to detect and classify the severity of anxiety and depression [23]. It comprises 2 parts, with 7 questions each, relating to anxiety and depressive symptoms, respectively. Each answer is rated from 0 to 3, for a total score out of 42 and anxiety and depression scores out of 21 each. The French version of HADS has good reliability and discriminant validity; the internal consistency of the two scales is good. For anxiety, Cronbach’s alpha is 0.81, and for depression, 0.78 [24]. A meta-analysis showed a sensitivity of 82% and specificity of 74% [25]. In addition, in 2002, Bjelland et al. reported good sensitivity and specificity in the detection of anxiety and depressive disorders, in particular outside psychiatric settings [26]. In the present study, the main endpoint was anxiety symptoms, assessed on the French HADS. The Anxiety dimensions were rated on 3 levels: no symptoms (score ≤ 7), doubtful (8–10), and certain (≥ 11). A cut-off at 8 points defined clinical signs suggestive of anxiety disorder. A score of 11 was considered a threshold for each subcategory and was previously reported to be valid for defining the presence or absence of anxiety or depressive symptomatology [25,26].

PCL-5 scale to assess post-traumatic stress disorder.

The PCL-5 scale (PTSD Checklist for DSM-5) was created by Weathers et al. in 2013 and translated into French by Desbiendras [27,28]. It is a 20-item self-report questionnaire that assesses PTSD symptoms according to DSM-5 criteria. It is used to screen for PTSD, establish a provisional diagnosis of PTSD, and assess changes in symptomatology. Items are scored from 0 (“not at all”) to 4 (“extremely”). A cut-off value of 38 suggests the presence of PTSD [27].

Calculation of scores for each validated scale identified proven signs of these pathologies when above the threshold. A message interpreting the score with advice for medical evaluation and management was included at the end of the questionnaire.

In the context of the COVID-19 health crisis, socio-occupational and medical factors associated with these mental pathologies were sought on the basis of questions included in the questionnaire:Socio-occupational factors: age, gender, family situation, job, working hours, seniority.Experience of the COVID crisis:
○Difficulties in reconciling family and working life;○Feelings of inadequate protection against COVID-19 infection risk;○Feelings of concern about the risk of transmission of COVID-19 for residents, for relatives, for their own health, for colleagues and superiors;○Feelings of difficulties in accompanying residents with COVID-19 or their families
–due to lack of equipment;–due to lack of communication within the facility;–due to a feeling of dehumanization of care–due to lack of time;–due to a feeling of powerlessness;○Feelings of difficulties in accompanying residents with COVID-19 or their families;○Confrontation with the deaths of COVID-19 residents and emotional effects;○Feeling of a traumatic work-related event during the health crisis.Medical factors:○History of anxiety disorder;○History of depressive syndrome;○Psychotropic treatment;○Work stoppage in the last 12 months;○Duration of the work stoppage;○Feeling of a traumatic work-related event during the health crisis.

Perceived stress related to personal and educational life was assessed on a visual analog scale (VAS) [29], with a cut-off at 7 points defining clinical signs of stress.

The collection of information on vaccination in our study was not foreseen in the methodology developed in July 2020 due to a lack of information on the availability of a vaccine for staff in 2021.

### 2.3. Analysis

Qualitative evaluation comprised transverse and comparative analysis of thematic content. Quantitative evaluation included descriptive analysis of sociodemographic, occupational, and medical characteristics. Univariate analysis assessed the association between anxiety, depression, and PTSD symptoms and sociodemographic, occupational, and medical factors. Chi^2^ and Fisher tests were applied as appropriate. The significance threshold was set at 5%. As the prevalence of the events (anxiety, depression, or PTSD) were high, odds ratios would not provide a good estimate of relative risk, rather, the log-linked binomial model was applied, using the PROC GENMOD procedure in the SAS statistical package ((SAS Institute Inc., SAS Campus Drive, Cary, NC 27513, USA, version 9.4) with the DIST = BINOMIAL and LINK = LOG options. Multivariable analyses were performed for each outcome (anxiety, depression, PTSD), variables with a *p*-value ≤ 0.05 in the univariate step were included in a multivariate model by a descending procedure, and variables with *p*-values < 0.05 remained in the model.

## 3. Results

### 3.1. Pilot Work: Qualitative Assessment

#### 3.1.1. Theme 1: Questionnaire Acceptability

Half of the respondents were positive about the acceptability of the questionnaire and underlined its interest.



*“I found the questionnaire easy to fill out, and the questions were quite clear.”*


*“It was the first time that we were concerned about nursing home staff. I have never seen studies like this in social and medical establishments.”*


*“I think it can be a good way indeed to reach more people.”*



#### 3.1.2. Theme 2: Questionnaire Administration Conditions

The methods of distributing the questionnaire online via a poster with QR codes or in paper format varied according to the institution. Anonymous distribution and the setting up of collection boxes facilitated participation and thus adherence. However, placing the ballot box in the office of a member of the hierarchy can be a hindrance to participation.



*“So there were two modes of distribution. Indeed, the questionnaires were made available as paper questionnaires at the entrance where all the professionals pass by, there was also a box where they could be left.” “There were a few stapled sheets of paper, placed on the supervisor’s desk, I believe, which were handed out during relief. (…) And it was to be filled out and put in a cardboard box.”*


*“Online it was proposed to people who have direct access to the Internet here.”*



#### 3.1.3. Theme 3: Expectations Regarding Psychological Care and Implementation of Preventive Actions

Some participants emphasized the individual usefulness of this questionnaire as a source of questioning and expression of the experience of working conditions and the state of psychological health, and the collective usefulness of this questionnaire for collecting collective data with a view to proposing preventive actions.



*“We have difficulties in our job. It’s an overload of work… a lack of staffing…”*


*“In this moment of crisis, it was still nice to know that we were being listened to through a questionnaire. It was to be heard but it was also to write down our feelings on paper.”*


*“It’s like I said: the questionnaire is fine, but there must be things put in place behind it.”*


*“When you give meaning to an action, the result is different afterwards and that’s why we have to follow up this survey.”*



According to this qualitative assessment, the questionnaire was well accepted by the participants and met their wishes for prevention action.

### 3.2. Quantitative Results

#### 3.2.1. Sociodemographic, Occupational, and Medical Characteristics

Out of 1117 eligible subjects, 386 working in 12 nursing homes participated in the study (participation rate: 34.5%). As shown in Table 1, 373 of the completed questionnaires could be included in the analysis. The sample size data was 373. The sample was 82% female (N = 306), and 70.7% of the participants (N = 263) lived in couples. More than two-thirds were caregivers. More than three-quarters worked 35 h per week or more. Mean age was 41.7 years [SD = 11.7]. Mean seniority was 11.9 years [SD = 9.3 years], and mean length of service in the present nursing home was 9.6 years [SD = 9]. As shown in Table 1, the prevalence was higher for anxiety symptoms (21%) than for depressive symptoms (10%) or PTSD (7%). A total of 8% of the subjects interviewed were taking a psychotropic treatment, two-thirds for less than a year. More than half of the respondents had taken time off work, half of whom for 2 weeks or more; 18% indicated a reason for being off work related to mental pathology. Nearly three-quarters of the employees interviewed expressed difficulties in caring for residents infected with COVID-19 and their families, related to a feeling of lack of time, powerlessness, lack of personnel, a perceived dehumanization of care, and, less frequently, perceived lack of equipment and communication within the facility (Figure 1).

#### 3.2.2. Factors Associated with Anxiety Symptoms

As shown in Table 2, on univariate analysis, anxiety symptoms were associated with:Difficulties in balancing family and working life (PR = 2.7 [1.7–4.3]).Medical factors:–History of anxiety disorder (PR = 3.1 [2.2–4.5]);–History of depressive disorder (PR = 2.8 [1.9–4.1]);–Psychological/psychiatric care (PR = 2.3 [1.5–3.6]);–Work stoppage related to mental health condition (PR = 2.0 [1.1–3.6]).Occupational factors:–Feeling of inadequate protection against COVID-19 infection (PR = 1.8 [1.2–2.6]);–Concern about the risk of transmission of COVID-19 for one’s own health (PR = 1.6 [1.1–2.3]), or for one’s colleagues or superiors (PR = 1.6 [1.1–2.4]);–Feeling of difficulty in accompanying residents infected with COVID-19 or their families due to lack of equipment (PR = 1.2 [0.5–2.6]), lack of time (PR = 6.7 [0.9–47.5]), or perceived powerlessness (PR = 2.8 [0.7–10.5]);–High level of occupational stress (PR = 8.7 [3.6–20.9]) or personal stress (PR = 5 [3.1–8.0]);–Perceived traumatic event at work during the health crisis (PR = 2.3 [1.5–3.6]) or at home during the health crisis (PR = 1.8 [1.2–2.7]).


In contrast, anxiety symptoms were not significantly associated with gender (*p* = 0.8) and age (*p* = 0.4) and family situation (*p* = 0.3).

As shown in Table 3, on multivariate analysis, anxiety symptoms remained associated with
–Feeling of inadequate protection against COVID-19 infection (PR = 1.4 [1.2–2.6]);–History of anxiety disorder (PR = 1.4 [1.2–1.6]);–Psychological/psychiatric care (PR = 1.4 [1.2–1.6]);–Level of high occupational stress in the last 3 months (PR = 5.0 [2.0–12.5]).–Level of high personal stress in the last 3 months (PR = 2.0 [1.2–3.4])

#### 3.2.3. Factors Associated with Depressive Symptoms

As shown in Table 2, on univariate analysis, depressive symptoms were associated with:Difficulties in reconciling family and working life (PR = 2.9 [1.3–6.2]).medical factors:–History of anxiety disorder (PR = 2.5 [1.3–4.6]);–History of depressive disorder (PR = 3.4 [1.8–6.4]).Occupational factors:–Concern about the risk of transmission of COVID-19 to residents (PR = 2.7 [0.9–7.5]), or to colleagues or superiors (PR = 2.0 [1.1–3.8]);–High level of occupational stress (PR = 5.4 [2.0–15.1]) or personal stress (PR = 4.8 [2.1–10.7]).

In contrast, depressive symptoms were not significantly associated with gender (*p* = 0.8) and age (*p* = 0.8) and family situation (*p* = 0.9).

As shown in Table 3, on multivariate analysis, depressive symptoms remained associated with:–History of depressive disorder (PR = 2.5 [1.3–4.6]);–Level of high occupational stress in the last 3 months (PR = 4.9 [1.8–13.6]).

#### 3.2.4. Factors Associated with Signs of PTSD

As shown in Table 2, on univariate analysis, signs of PTSD were associated with:Difficulties in reconciling family and working life (PR = 3.0 [1.2–7.2]).Medical factors:–History of anxiety disorders (PR = 3 [1.5–6.2]);–History of depressive disorders (PR = 5.3 [2.6–10.7]).occupational factors:–Perceived difficulty in accompanying residents infected with COVID-19 or their families, linked to lack of communication within the structure (PR = 3.6 [0.9–13.1]), or to perceived dehumanization of care (PR = 7.4 [1.1–54.9]);–High level of occupational stress (PR = 2.1 [0.8–5.2]) or personal stress (PR = 3.8 [1.6–8.9]);–Perceived traumatic event in the personal environment during the health crisis (PR = 2.5 [1.2–5.1]).

In contrast, depressive signs of PTSD were not significantly associated with gender (*p* = 0.7) and age (*p* = 0.8) and family situation (*p* = 0.7).

As shown in Table 3, on multivariate analysis, signs of PTSD remained associated with:–History of depressive disorder (PR = 4.9 [2.4–9.8]);–Experiencing a traumatic event in personal life (PR = 2.2 [1.1–4.4]).

Although the results of the univariate analysis show an association between difficulties in reconciling family/working life imbalance and anxiety symptoms, depressive symptoms, or PTSD, the association is not maintained after multivariate analysis. On the other hand, medical factors (such as a history of anxiety or depressive symptoms) and occupational factors (such as high occupational stress) remained associated with these pathologies after multivariate analysis in nursing home staff involved in the COVID-19 health crisis.

## 4. Discussion

During the pandemic, changes in working conditions in nursing homes and long-term care facilities have affected residents, their families, and health and social workers [30].

The use of a self-administered questionnaire to identify workers with impaired mental health is not a standard medical practice in occupational medicine. In a context of increasing workload for nursing home staff and a shortage of occupational physicians, this study provides a qualitative and quantitative assessment of a tool to detect mental health issues using a self-report questionnaire.

These screening methods could make it possible to identify mental disorders at an early stage with the potential effect of improving mental health and reducing the number of work stoppages.

Previous studies have shown the value of early detection of mental illness in medical management and have shown that early identification of mental illness without appropriate medical management can be associated with increased injury [31,32]. The qualitative evaluation of this tool for screening for mental disorders underlines the interest, the acceptability of the modalities of taking the self-administered questionnaire, and the need to associate psychological support for the nursing home staff. The implementation literature emphasizes that it is important to create a positive climate for implementation [33]. Interest and acceptability are favorable criteria for the implementation of this tool for the identification of mental disorders.

The study showed the individual and collective interest in identifying employees with signs suggestive of anxiety, depression, and PTSD. It also underlined the personal and occupational factors associated with these pathologies in order to identify possible areas for prevention. According to the qualitative assessment, more than half of the participants accepted the questionnaire and found it interesting to complete.

Studies that examined mental health outcomes in healthcare workers before and during the COVID-19 outbreak found a significant increase in reported anxiety symptoms in the outbreak period compared to the non-outbreak period [34,35]. In the present study, the prevalence of anxiety disorder in nursing home staff was 21%, similar to previous reports. According to data from the French Public Health Agency, the prevalence of anxiety in the general French adult population was 23% in the first COVID wave, compared to 13.5% in 2017 [36]. A cross-sectional survey of 1422 Spanish health workers evaluated the prevalence of anxiety at 20.7% [37]. A systematic review of the literature with meta-analysis estimated the prevalence of anxiety in caregivers during COVID-19 as 25% [38]. A systematic review with meta-regression found a prevalence of anxiety of 25.8% in hospital caregivers managing patients affected by COVID-19 [39]. Another systematic review, including 12 studies, found a pooled anxiety prevalence of 23.2% in healthcare workers [40].

The prevalence of depressive disorder in the present study was consistent with data from the French Public Health Agency in 2011 in a sample of employees (7.11%) before the COVID-19 health crisis [41]. However, these findings were lower than those of studies focusing exclusively on healthcare workers. The systematic review of the literature with meta-analysis found a prevalence of depression of 22.8% in healthcare workers during the COVID-19 pandemic, however, the studies used a variety of self-report anxiety scales, and the use of some tests was associated with a significantly higher prevalence of anxiety than others [38]. A systematic review with meta-regression found a prevalence of 24.3% for depression in hospital healthcare workers caring for patients affected by COVID-19 [39]. A systematic review including 10 studies found a pooled depression prevalence of 22.8% [40]. A cross-sectional survey on 1422 Spanish healthcare workers evaluated the prevalence of depression at 15.2% [37]. The difference in the assessment of the prevalence of depression in our study may be because our sample included healthcare workers and non-healthcare workers and because working conditions differ between nursing homes for the elderly and hospitals.

The data collected by the French Public Health Agency in French employees before the beginning of the health crisis showed a prevalence of 2.3% in 2011, compared to 5–12% in the general American population [41,42]. Previous studies highlighted an important impact of the COVID-19 pandemic on the mental health of healthcare workers [43]. The prevalence of clinically relevant trauma-related symptoms ranged from 7.4% to 35% [44,45]. Our study estimated the prevalence of PTSD at 7%, lower than in a previous report of 13.2% in Chinese healthcare workers [46]. A multicenter observational cohort study of 9138 healthcare workers, carried out in a convenience sample of 18 healthcare institutions from 6 Autonomous Communities in Spain, showed a 22% prevalence of PTSD [47]. The differences in these results could be explained by different contagion rates and pressure on healthcare systems, different incidences of risk factors, and differences in access to psychological support [48]. Not all nursing homes experienced the crisis in the same way; for example, some had many patients infected with COVID but very few deaths, which may limit the psychological impact on staff. Strong support and team cohesion in the nursing homes may have reduced the effect of the stress induced by the health crisis.

In our study, anxiety symptoms, depressive symptoms were not significantly associated with gender, which is inconsistent with the data in the literature [44,49,50]. Moreover, anxiety symptoms and depressive symptoms were not significantly associated with age, whereas differences by age were reported, with younger Health Care Workers experiencing higher levels of anxiety and depressive symptoms compared to older age groups in the literature [14,51,52].

This study revealed factors associated with anxiety, depression, and PTSD and identified potential areas for prevention.

According to our study, the medical factors associated with anxiety disorders, depressive disorders, and PTSD were history of anxiety disorder and depressive disorder. These results are similar to those of previous studies. A multicenter, observational cohort study of 9138 healthcare workers reported that the higher the number of prior lifetime mental disorders, the more likely the presence of any current disorder [47]. Şahin et al. conducted a cross-sectional study of 939 healthcare workers during the COVID-19 pandemic in Turkey between 23 April and 23 May 2020. They showed that depression and anxiety symptoms were significantly greater in individuals with a history of psychiatric illness [53].

Our study also highlighted an association between exposure to modifiable occupational factors and the presence of anxiety, depression, or PTSD. There was an association between anxiety symptomatology and feelings of lack of time and of helplessness in caring for residents infected with COVID-19 or their families in this health crisis. Previous studies reported that nursing homes had difficulty organizing and managing material and human resources to respond to the pandemic [54]. This reorganization required nursing homes to make decisions on the safety of residents and employees, including protocols for isolation and evacuation and for screening people with symptoms or positive COVID-19 diagnosis, which increased stress levels.

Our study also highlighted an association between staff’s concern about transmitting infection, difficulties in supporting residents and their families, and mental illness. These results corroborate those in the literature. Previous studies showed an association between mental disorders and the high mental burden to which workers are exposed, excessive working hours, fear of contagion for themselves and their loved ones, lack of job resources, and the high degree of suffering of patients and their families [17,55,56,57].

High levels of stress and difficulties in balancing family and working life are modifiable factors that could be used as a basis for prevention measures. These findings are consistent with the literature.

Previous studies described the high levels of stress and pressure affecting nursing home employees (uncertainty, hopelessness, excess workload, and role conflicts), with a significant impact on their mental health [58,59]. In a study of 152 front-line nursing home staff, White et al., underlined the emotional burden of caring for residents experiencing distress, illness, and death [58]. Some employees commented on the fear and stress associated with possibly being infected and infecting family members. Spoorthy et al., in a review of the literature, highlighted the increased stress of health care personnel confronted with a health crisis and its association with anxiety and depressive disorders [60]. The authors underlined the positive effects of social support and psychological accompaniment of personnel confronted with the health crisis. Training in better stress management and a participatory approach to improving personal/working life could be proposed to nursing home staff.

Moreover, in line with the literature, our qualitative findings strongly suggest establishing psychological support services for providing adequate professional care for nursing home staff [61,62].

The use of this tool for the longitudinal medical follow-up of employees would allow the early identification of personnel requiring psychological care. The data in the literature underline the importance of encouraging the implementation of psychological support for staff involved in the health crisis. Nursing home staff are eager to receive psychological care [17,63,64]. Nursing home managers could use these results to implement human resources to promote psychological support for staff.

### Study Strengths and Weaknesses

In this cross-sectional study, the identified factors were regarded as associated factors, which could be either causes or results of anxiety, depression disorders, and PTSD. The results should be generalized with caution since they were drawn from a non-randomized sample and the sample size was small, and the response rate was 34.5%. Furthermore, anxiety, depressive symptoms, and PTSD were identified not on clinical examination but on a validated scale. The collection method based on a self-administered questionnaire makes it possible to collect medical information and information on the experience of working conditions during a health crisis. The declarative nature of the information constitutes a potential bias in data classification. The controlled length of the questionnaire, its content, and presentation contributed to its completion.

Vaccination against COVID-19 may have had an effect on the prevalence of anxiety, depression, and Post-Traumatic Stress disorder in nursing home staff involved in the COVID-19 health crisis. According to Haddaden et al., a COVID-19 vaccination improved healthcare workers’ comfort and anxiety in caring for patients with COVID-19 and other illnesses likely explained by the high efficacy of vaccination in preventing symptomatic infection and critical illness [63]. The COVID-19 vaccination coverage of the sample was not known. However, the data from Santé Publique France, on 16 March 2021, showed that 20.8% of professionals working in retirement homes in the Loire had received a first dose of vaccine. It is estimated that 74.6% of the residents of retirement homes located in the Loire have received a first dose of the COVID-19 vaccine [64].

## 5. Conclusions

The questionnaire was well accepted by the participants and met their wishes for prevention actions. The study showed the interest in early detection of mental disorders by a self-report questionnaire in nursing home staff involved in the health crisis to optimize medical care. The survey showed that it is possible to encourage regular medical monitoring of professionals by using this kind of questionnaire. Moreover, the study highlighted possible lines of prevention based on better stress management and better reconciliation of family and working life for personnel involved in the health crisis. Future studies should clarify the long-term effects of the COVID-19 pandemic on the mental health of healthcare workers in nursing homes for the elderly.

## Figures and Tables

**Figure 1 ijerph-19-04061-f001:**
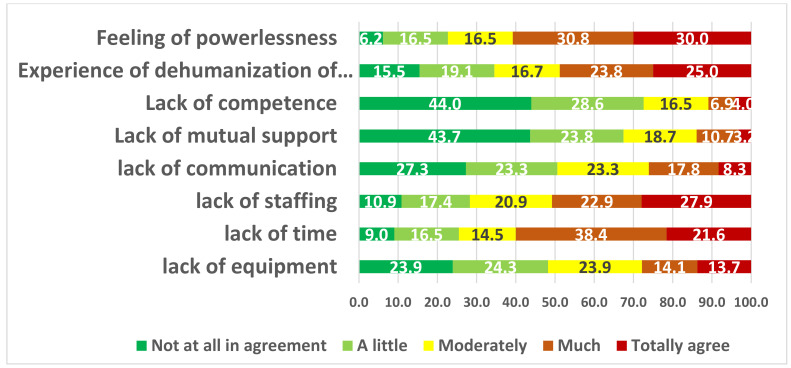
Causes of difficulties accompanying residents infected with COVID-19 and their families.

**Table 1 ijerph-19-04061-t001:** Socio-occupational and medical characteristics.

Medical Factors		*n*	%	Occupational Factors		*n*	%
**History of anxiety disorder (*n* = 366)**	**No**	298	81.4	**Difficulties in reconciling family and working life (*n* = 367)**	**No**	169	46.1
	**Yes**	68	18.6		**Yes**	198	53.9
**History of depressive syndrome (*n* = 367)**	**No**	325	88.6	**Feeling of inadequate protection against COVID-19 infectious risk (*n* = 366)**	**No**	219	59.8
	**Yes**	42	11.4		**Yes**	147	40.2
**Proven symptoms of anxiety (*n* = 373)**	**No**	294	78.8	**Feeling of concern about the risk of transmission of COVID-19 (*n* = 365)**	**No**	23	6.3
	**Yes**	79	21.2		**Yes**	342	93.7
**Proven symptoms of depression (*n* = 373)**	**No**	337	90.3	**For residents (*n* = 370)**	**No**	94	25.4
	**Yes**	36	9.7		**Yes**	276	74.6
**Symptoms of post-traumatic stress disorder (*n* = 373)**	**No**	346	92.8	**For relatives (*n* = 370)**	**No**	54	14.6
	**Yes**	27	7.2		**Yes**	316	85.4
**Taking a psychotropic treatment (*n* = 363)**	**No**	332	91.5	**For your own health (*n* = 370)**	**No**	186	50.3
	**Yes**	31	8.5		**Yes**	184	49.7
**If taking a psychotropic treatment, since when? (*n* = 31)**	**Less than 3 months**	3	9.7	**For colleagues and superiors (*n* = 392)**	**No**	223	60.4
	**Between 3 and 6 months**	5	16.1		**Yes**	146	39.6
	**Between 6 months and 1 year**	13	41.9	**Feeling of difficulties in accompanying residents with COVID-19 or their families** **(*n* = 366)**	**No**	103	28.1
	**More than 1 year**	10	32.3		**Yes**	263	71.9
**Work stoppage in the last 12 months (*n* = 365)**	**No**	177	48.5	**Have you been confronted with the deaths of COVID-19 residents? (*n* = 370)**	**No**	103	27.8
	**Yes**	188	51.5		**Yes**	267	72.2
**If yes, what duration of work stoppage? (*n* = 183)**	**[1–7 days]**	51	27.9	**If Yes, were you emotionally affected? (*n* = 265)**	**No**	32	12.1
	**[8–14 days]**	42	23.0				
	**[15–29 days]**	55	30.0		**Yes**	233	87.9
	**More than 30 days**	35	19.1				

**Table 2 ijerph-19-04061-t002:** Factors associated with anxiety symptoms, depressive symptoms, and Post-Traumatic Stress Disorder after univariate analysis.

		Proven Symptoms of Anxiety						Proven Symptoms of Depression						Signs of Post-Traumatic Stress Disorder					
		Yes		No				Yes		No				Yes		No			
		*n*	%	*n*	%	PR	CI	*n*	%	*n*	%	PR	CI	*n*	%	*n*	%	PR	CI
**Difficulties in reconciling family and working life**	No	19	24.1	150	52.1	1 ****	-	8	22.9	161	48.5	1 **	-	6	22.2	163	47.9	1 **	-
	Yes	60	75.9	138	47.9	2.7	1.7–4.3	27	77.1	171	51.5	2.9	1.3–6.2	21	77.8	177	52.1	3.0	1.2–7.2
**Feeling of inadequate protection against COVID 19 infectious risk in the workplace**	No	36	45.6	183	63.8	1 ***	-	17	48.6	202	61.0	1	-	12	44.4	207	61.1	1	-
	Yes	43	54.4	104	36.2	1.8	1.2–2.6	18	51.4	129	39.0	1.6	0.8–2.9	15	55.6	132	38.9	1.9	0.9–3.9
**Feeling of concern about the risk of transmission of COVID-19 to residents**	No	14	17.7	80	27.5	1	-	4	11.1	90	27.0	1 *	-	3	11.1	91	26.5	1	-
	Yes	65	82.3	211	72.5	1.6	0.9–2.7	32	88.9	244	73.0	2.7	0.9–7.5	24	88.9	252	73.5	2.7	0.8–8.8
**Feeling of concern about the risk of transmission of COVID-19 for your own health**	No	31	39.2	155	53.3	1 *	-	16	44.4	170	50.9	1	-	11	40.7	175	51.0	1	-
	Yes	48	60.8	136	46.7	1.6	1.1–2.3	20	55.6	164	49.1	1.3	0.7–2.4	16	59.3	168	49.0	1.5	0.7–3.1
**Feeling of concern about the risk of transmission of COVID-19 for colleagues and superiors**	No	38	48.1	185	63.8	1 *	-	15	42.9	208	62.3	1 *	-	12	44.4	211	61.7	1	-
	Yes	41	51.9	105	36.2	1.6	1.1–2.4	20	57.1	126	37.7	2.0	1.1–3.8	15	55.6	131	38.3	1.9	0.9–4.0
**Feeling of difficulties in accompanying residents with COVID-19 or their families due to lack of equipment**	Not at all in agreement	12	20.7	49	24.9	1 *	-	4	13.8	57	25.2	1	-	4	18.2	57	24.5	1	-
	A little	10	17.2	52	26.4	0.8	0.4–1.8	7	24.1	55	24.3	1.7	0.5–5.6	6	27.3	56	24.0	1.5	0.4–5.0
	Moderately	12	20.7	49	24.9	1.0	0.5–2.0	6	20.7	55	24.3	1.5	0.4–5.1	3	13.6	58	24.9	0.8	0.2–3.2
	Much	16	27.6	20	10.1	2.2	1.2–4.2	7	24.1	29	12.8	3.0	0.9–9.4	7	31.8	29	12.4	3.0	0.9–9.4
	Totally agree	8	13.8	27	13.7	1.2	0.5–2.6	5	17.2	30	13.3	2.2	0.6–7.6	2	9.1	33	14.2	0.9	0.2–4.5
**Feeling of difficulties in accompanying residents with COVID-19 or their families due to lack of communication within the facility**	Not at all in agreement	17	28.8	52	26.8	1	-	9	30.0	60	26.9	1	-	3	13.0	66	28.7	1 *	-
	A little	13	23.7	46	23.7	0.9	0.5–1.7	3	10.0	56	25.1	0.4	0.1–1.4	3	13.0	56	24.3	1.2	0.2–5.6
	Moderately	16	22.2	43	22.2	1.1	0.6–2.0	10	33.3	49	22.0	1.3	0.6–3.0	10	43.5	49	21.3	3.9	1.1–13.5
	Much	10	18.0	35	18.0	0.9	0.5–1.8	7	23.3	38	17.0	1.2	0.5–3.0	7	30.5	38	16.5	3.6	0.9–13.1
	Totally agree	3	5.1	18	9.3	0.6	0.2–1.8	1	3.3	20	9.0	0.4	0.1–2.7	0	0	21	9.1	-	-
**Feeling of difficulties in accompanying residents with COVID-19 or their families due to a feeling of dehumanization of care**	Not at all in agreement	6	10.7	33	16.8	1	-	4	13.8	35	15.7	1	-	1	4.4	38	16.6	1 *	-
	A little	7	12.5	41	20.9	0.9	0.3–2.6	4	13.8	44	19.7	0.8	0.2–3.0	2	8.7	46	20.1	1.6	0.2–17.3
	Moderately	7	12.5	35	17.9	1.1	0.4–2.9	2	6.9	40	17.9	0.5	0.1–2.4	4	17.4	38	16.6	3.7	0.4–31.8
	Much	19	33.9	41	20.9	2.1	0.9–4.7	9	31.0	51	22.9	1.5	0.5–4.2	4	17.4	56	24.4	2.6	0.3–22.4
	Totally agree	17	30.4	46	23.5	1.8	0.8–4.1	10	34.5	53	23.8	1.5	0.5–4.6	12	52.2	51	22.3	7.4	1.1–54.9
**Feeling of difficulties in accompanying residents with COVID-19 or their families due to lack of time**	Not at all in agreement	1	1.7	22	11.2	1 *	-	1	3.3	22	9.8	1	-	2	8.7	21	9.1	1	-
	A little	6	10.3	36	18.3	3.3	0.4–25.7	4	13.3	38	16.9	2.2	0.2–18.5	2	8.7	40	17.2	0.5	0.1–3.6
	Moderately	6	10.3	31	15.7	3.7	0.5–29.0	3	10.0	34	15.1	1.9	0.2–16.9	3	13.0	34	14.7	0.9	0.2–5.2
	Much	29	50.0	69	35.0	6.8	1.0–47.4	14	46.7	84	37.3	3.3	0.5–23.7	10	43.5	88	37.9	1.2	0.3–5.0
	Totally agree	16	19.8	39	19.8	6.7	0.9–47.5	8	26.7	47	20.9	3.3	0.4–25.2	6	26.1	49	21.1	1.2	0.3–5.8
**Feeling of difficulties in accompanying residents with COVID-19 or their families due to a feeling of powerlessness**	Not at all in agreement	2	3.4	14	7.0	1 ***	-	0	0	16	6.9	-	-	0	0	16	6.7	-	
	A little	2	3.4	41	20.4	0.4	0.1–2.4	3	10.0	40	17.4	1	-	1	4.4	42	17.7		
	Moderately	6	10.2	37	18.4	1.1	0.3–5.0	3	10.0	40	17.4	1.0	0.2–4.7	2	8.7	41	17.3		
	Much	22	37.3	58	28.9	2.2	0.6–8.4	8	26.7	72	31.3	1.4	0.4–5.1	5	21.7	75	31.7		
	Totally agree	27	45.8	51	25.4	2.8	0.7–10.5	16	53.3	62	27.0	2.9	0.9–9.5	15	65.2	63	26.6		
**History of anxiety disorder**	No	46	58.2	252	87.8	1 ****	-	23	63.9	275	83.3	1 **	-	16	59.3	282	83.2	1 **	-
	Yes	33	41.8	35	12.2	3.1	2.2–4.5	13	36.1	55	16.7	2.5	1.3–4.6	11	40.7	57	16.8	3.0	1.5–6.2
**History of depressive disorder**	No	58	73.4	267	92.7	1 ****	-	25	69.4	300	90.6	1 ***	-	16	59.3	309	90.9	1 ****	-
	Yes	21	26.6	21	7.3	2.8	1.9–4.1	11	30.6	31	9.4	3.4	1.8–6.4	11	40.7	31	9.1	5.3	2.6–10.7
**Psychotropic treatment**	No	58	73.4	269	93.4	1 ****	-	25	69.4	302	91.2	1 ***	-	18	66.7	309	90.9	1 ***	-
	Yes	21	26.6	19	6.6	3.0	2.0–4.3	11	30.6	29	8.8	3.6	1.9–6.7	9	33.3	31	9.1	4.1	2.0–8.5
**Psychological/psychiatric care**	No	65	82.3	267	94.0	1 **	-	29	80.6	303	92.7	1 *	-	19	70.4	313	93.1	1 ****	-
	Yes	14	17.7	17	6.0	2.3	1.5–3.6	7	19.4	24	7.3	2.6	1.2–5.4	8	29.6	23	6.9	4.5	2.2–9.4
**Work stoppage related to mental health condition**	No	25	69.4	126	84.6	1 *	-	9	50.0	142	85.0	1 ***	-	11	73.3	140	82.3	1	-
	Yes	11	30.6	23	15.4	2.0	1.1–3.6	9	50.0	25	15.0	4.4	1.9–10.3	4	26.7	30	17.7	1.6	0.5–4.8
**Level of occupational stress in the last 3 months**	light	5	6.5	97	34.6	1 ****	-	4	11.4	98	30.4	1 ****	-	6	23.1	96	29.0	1 *	-
	moderate	20	26.0	113	40.4	3.1	1.2–7.9	5	14.3	128	39.8	0.9	0.3–3.5	5	19.2	128	38.7	0.6	0.2–2.0
	high	52	67.5	70	25.0	8.7	3.6–20.9	26	74.3	96	29.8	5.4	2.0–15.1	15	57.7	107	32.3	2.1	0.8–5.2
**Level of personal stress in the last 3 months**	light	19	24.4	159	56.0	1 ****	-	8	22.8	170	52.0	1 ***	-	8	29.6	170	50.8	1 ***	-
	moderate	22	28.2	92	32.4	1.8	1.0–3.2	12	34.3	102	31.2	2.3	0.9–5.5	7	25.9	107	31.9	1.4	0.5–3.7
	high	37	47.4	33	11.6	5.0	3.1–8.0	15	42.9	55	16.8	4.8	2.1–10.7	12	44.4	58	17.3	3.8	1.6–8.9
**Feeling of a traumatic work-related event during the health crisis**	No	27	34.2	174	60.6	1 ****	-	18	50.0	183	55.4	1	-	11	40.7	190	56.0	1	-
	Yes	52	65.8	113	39.4	2.3	1.5–3.6	18	50.0	147	44.6	1.2	0.7–2.3	16	59.3	149	44.0	1.8	0.8–3.7
**Feeling of a traumatic event in personal life during the health crisis**	No	50	63.3	227	79.1	1 ***	-	27	75.0	250	75.8	1	-	15	55.6	262	77.3	1 *	-
	Yes	29	36.7	60	20.9	1.8	1.2–2.7	9	25.0	80	24.2	1.1	0.5–2.1	12	44.4	77	22.7	2.5	1.2–5.1

PR: Prevalence Ratio; CI: Confidence Interval; *p*-value: *: *p* < 0.05 **: 0.01 < *p* ≤ 0.05; ***: 0.001 < *p* ≤ 0.01; **** *p* ≤ 0.0001.

**Table 3 ijerph-19-04061-t003:** Factors associated with anxiety symptoms, depressive symptoms, and Post-Traumatic Stress Disorder after multivariate analysis.

Experiencing a Traumatic Event in Personal Life (PR = 1.4 [1.2–1.6])		Proven Symptoms of Anxiety		Proven Symptoms of Depression		Signs of Post-Traumatic Stress Disorder	
		PR adj	95% CI	PR adj	95% CI	PR adj	95% CI
**Feeling of inadequate protection against COVID 19 infectious risk in the workplace**	No	1 ****	-	/	/	/	/
	Yes	1.4	1.2–1.6	/	/	/	/
**History of anxiety disorder**	No	1 ****	-	/	/	/	/
	Yes	1.4	1.2–1.6	/	/	/	/
**History of depressive disorder**	No	/	/	1 ****	-	1 ****	-
	Yes	/	/	2.5	1.3–4.6	4.9	2.4–9.8
**Psychological/psychiatric care**	No	1 ****	-	/	/	/	/
	Yes	1.4	1.2–1.6	/	/	/	/
**Level of occupational stress in the last 3 months**	light	1 ****	-	1 ****	-	/	/
	moderate	2.6	1.1–6.8	0.9	0.3–3.4		
	high	5.0	2.0–12.5	4.9	1.8–13.6	/	/
**Level of personal stress in the last 3 months**	light	1 ****	-	/	/	/	/
	moderate	1.1	0.6–1.9	/	/	/	/
	high	2.0	1.2–3.4	/	/	/	/
**Experiencing a traumatic event in personal life during the health crisis**	No	1 ****	-	/	/	1 *	-
	Yes	1.4	1.2–1.6	/	/	2.2	1.1–4.4

PR adj: adjusted Prevalence Ratio CI: Confidence Interval; *: *p* < 0.05; *p*-value: **** *p* ≤ 0.0001; variables with a *p*-value ≤ 0.05 in the univariate step were included in a multivariate model by a descending procedure and variables with *p*-values < 0.05 remained in the model.

## Data Availability

The data presented in this study are available on request from the corresponding author. The data are not publicly available due to the confidentiality of participants.

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
