# Peer review of "COVID-19 Health Crisis Workloads and Screening for Psychological Impact in Nursing Home Staff: A Qualitative and Quantitative Survey"

_ijerph, 2022, doi:10.3390/ijerph19074061_

Round 1
Reviewer 1 Report
The topic of this paper focuses on the psychological problems of nursing home staff, which is very valuable. However, the paper has the following problems:
- The sample size of this study is not clear. "A multicenter observational study was conducted in 1117 nursing home staff in France" is mentioned in the abstract, but "out of 1117 eligible subjects, 386 working in 12 nursing home participated in the study" "373 of the completed questions could be included in the analysis" is mentioned in lines 157-158 of the paper.Should the actual sample size be 373? Excluding 386 staff, who are the rest of the 1117? The sample size data of answers to various questions in Table 1 are 365, 366 and 367 respectively, which is in contradiction with 373 mentioned above. The expression of this part is very confusing.
- The overall research design is relatively simple. The quantitative and qualitative analysis part only lists the analysis results,. What is the contribution of the interview? It is not clearly stated in the paper, and the authors did not mention the interview findings in the discussion. Through the data analysis results, we can see that there are a large number of relevant factors, but which of these factors is more important or more common? Analyzing these factors may be more valuable to practical work.
- Although the research findings are summarized, what is the application value of these findings? How should we deal with so many psychological problems of nursing homestaff in practice? It is not mentioned in the paper.
Author Response
We thank the three reviewers for their suggestions of modifications which contribute to improve the article
REVIEWER 1
- The sample size of this study is not clear. "A multicenter observational study was conducted in 1117 nursing home staff in France" is mentioned in the abstract, but "out of 1117 eligible subjects, 386 working in 12 nursing home participated in the study" "373 of the completed questions could be included in the analysis" is mentioned in lines 157-158 of the paper.Should the actual sample size be 373? Excluding 386 staff, who are the rest of the 1117? The sample size data of answers to various questions in Table 1 are 365, 366 and 367 respectively, which is in contradiction with 373 mentioned above. The expression of this part is very confusing.
- Response to reviewer 1:
The abstract has been modified to specify the number of employees in the eligible population (n=1117), the number of employees who completed the questionnaire (n=386) and the number of questionnaires included in the analysis (n=373).
“A multicenter observational study was conducted in 12 nursing homes in France with 1117 nursing home staff eligible. The participation rate was 34.5% (N=386). Data from 373 questionnaires were included in the analysis.”
- The overall research design is relatively simple. The quantitative and qualitative analysis part only lists the analysis results,. What is the contribution of the interview? It is not clearly stated in the paper, and the authors did not mention the interview findings in the discussion. Through the data analysis results, we can see that there are a large number of relevant factors, but which of these factors is more important or more common? Analyzing these factors may be more valuable to practical work.
- Response to reviewer 1:
The authors summarized the main results of the univariate analysis in a single table (Table 2) to clarify the results.
Moreover, the authors have outlined in the discussion, the interview finding.
“These screening methods could make it possible to identify mental disorder at an early stage with a potential effect of improving mental health and reducing the number of work stoppages.
Previous studies have shown the value of early detection of mental illness in medical management and have shown that early identification of mental illness without appropriate medical management can be associated with increased injury [31,32]. The qualitative evaluation of this tool for screening for mental disorder underlines the interest, the acceptability of the modalities of taking the self-administered questionnaire and the need to associate psychological support for the nursing-home staff. The implementation literature emphasizes that it is important to create a positive climate for implementation [33]. Interest and acceptability are favorable criteria for the implementation of this tool for the identification of mental disorders.
Moreover, in line with the literature, our qualitative findings strongly suggest establishing psychological support services for providing adequate professional care for nursing homes staff [63,64].”
(31) Auerbach, R. P.; Mortier, P.; Bruffaerts, R.; Alonso, J.; Benjet, C.; Cuijpers, P.; Demyttenaere, K.; Ebert, D. D.; Green, J. G.; Hasking, P.; Murray, E.; Nock, M. K.; Pinder-Amaker, S.; Sampson, N. A.; Stein, D. J.; Vilagut, G.; Zaslavsky, A. M.; Kessler, R. C. WHO World Mental Health Surveys International College Student Project: Prevalence and Distribution of Mental Disorders. Journal of Abnormal Psychology 2018, 127 (7), 623–638. https://doi.org/10.1037/abn0000362.
(32) Pelissier, C.; Viale, M.; Berthelot, P.; Poizat, B.; Massoubre, C.; Tiffet, T.; Fontana, L. Factors Associated with Psychological Distress in French Medical Students during the COVID-19 Health Crisis: A Cross-Sectional Study. International Journal of Environmental Research and Public Health 2021, 18 (24), 12951. https://doi.org/10.3390/ijerph182412951.
(33) Kirk, J. W.; Sivertsen, D. M.; Petersen, J.; Nilsen, P.; Petersen, H. V. Barriers and Facilitators for Implementing a New Screening Tool in an Emergency Department: A Qualitative Study Applying the Theoretical Domains Framework. Journal of Clinical Nursing 2016, 25 (19–20), 2786–2797.
(63) Caillet, A.; Coste, C.; Sanchez, R.; Allaouchiche, B. Psychological Impact of COVID-19 on ICU Caregivers. Anaesthesia Critical Care & Pain Medicine 2020, 39 (6), 717–722. https://doi.org/10.1016/j.accpm.2020.08.006.
(64) Conti, C.; Fontanesi, L.; Lanzara, R.; Rosa, I.; Porcelli, P. Fragile Heroes. The Psychological Impact of the COVID-19 Pandemic on Health-Care Workers in Italy. PLoS One 2020, e0242538–e0242538.
(65) Tan, B. Y. Q.; Chew, N. W. S.; Lee, G. K. H.; Jing, M.; Goh, Y.; Yeo, L. L. L.; Zhang, K.; Chin, H.-K.; Ahmad, A.; Khan, F. A.; Shanmugam, G. N.; Chan, B. P. L.; Sunny, S.; Chandra, B.; Ong, J. J. Y.; Paliwal, P. R.; Wong, L. Y. H.; Sagayanathan, R.; Chen, J. T.; Ying Ng, A. Y.; Teoh, H. L.; Ho, C. S.; Ho, R. C.; Sharma, V. K. Psychological Impact of the COVID-19 Pandemic on Health Care Workers in Singapore. Annals of Internal Medicine 2020. https://doi.org/10.7326/M20-1083.
- Although the research findings are summarized, what is the application value of these findings? How should we deal with so many psychological problems of nursing homestaff in practice? It is not mentioned in the paper.
- Response to reviewer 1: The authors have specified in the discussion the modalities of psychological support for nursing home staff.
“The use of this tool for the longitudinal medical follow-up of employees would allow early identification of personnel requiring psychological care. The data in the literature underline the importance of encouraging the implementation of psychological support for staff involved in the health crisis. Nursing home staff are eager to receive psychological care [17,63,65]. Nursing home managers could use these results to implement human resources to promote psychological support for staff.”
(17) Cai, H.; Tu, B.; Ma, J.; Chen, L.; Fu, L.; Jiang, Y.; Zhuang, Q. Psychological Impact and Coping Strategies of Frontline Medical Staff in Hunan Between January and March 2020 During the Outbreak of Coronavirus Disease 2019 (COVID‑19) in Hubei, China. Med Sci Monit 2020, 26. https://doi.org/10.12659/MSM.924171.
(64) Conti, C.; Fontanesi, L.; Lanzara, R.; Rosa, I.; Porcelli, P. Fragile Heroes. The Psychological Impact of the COVID-19 Pandemic on Health-Care Workers in Italy. PLoS One 2020, e0242538–e0242538.
(65) Tan, B. Y. Q.; Chew, N. W. S.; Lee, G. K. H.; Jing, M.; Goh, Y.; Yeo, L. L. L.; Zhang, K.; Chin, H.-K.; Ahmad, A.; Khan, F. A.; Shanmugam, G. N.; Chan, B. P. L.; Sunny, S.; Chandra, B.; Ong, J. J. Y.; Paliwal, P. R.; Wong, L. Y. H.; Sagayanathan, R.; Chen, J. T.; Ying Ng, A. Y.; Teoh, H. L.; Ho, C. S.; Ho, R. C.; Sharma, V. K. Psychological Impact of the COVID-19 Pandemic on Health Care Workers in Singapore. Annals of Internal Medicine 2020. https://doi.org/10.7326/M20-1083.
Reviewer 2 Report
Dear authors,
Thank you very much for the opportunity to review this review. I read the work with great interest. It carries a lot of useful information, however, before making a decision, many inaccuracies should be clarified:
Generally:
1) Unification of the nomenclature "COVID-19" and SARS-CoV-2.
2) The bibliography requires correction and adaptation to the requirements of the journal.
3) Is the vaccination profile of the study participants known? Please describe this in the methodology section.
Admission
1) In the initial stage of the introduction, the authors provide statistical data without any information about the source - please complete the bibliography
2) At the end of the introduction, the research hypotheses on which the research was created with the source given should be presented
Results
1) In my opinion, to be removed completely section 3.2. this will also translate to a change of title and targets with the deletion of information on qualitative measurements.
2) Why did the authors choose PR instead of OR?
3) Figure 1 - very poor quality, correction needed.
Discussion
1) The discussion does not include a complete discussion of the influence of sociodemographic variables on the mental state, including age and gender, with a discussion of the results with world data.
2) I reiterate my question about the vaccination profile - if so, discuss it, if not add to the test limits as research shows that vaccination can have an impact on mental health.
3) The authors present the results from other countries, periods of the pandemic and before - was the impact of direct combat with the pandemic on health assessed? If so please discuss if not re-restrictions.
Author Response
We thank the three reviewers for their suggestions of modifications which contribute to improve the article
Reviewer 2
Generally:
1) Unification of the nomenclature "COVID-19" and SARS-CoV-2.
- Response to reviewer 2: In order to unify the nomenclature, SARS-CoV-2 has been replaced by COVID-9 in the introduction.
2) The bibliography requires correction and adaptation to the requirements of the journal.
- Response to reviewer 2:
Response to reviewer 2: The author have modified the bibliography to satisfy to the requirements of the journal (ASC style)
3) Is the vaccination profile of the study participants known? Please describe this in the methodology section.
- Response to reviewer 2:
It was not possible to obtain vaccination coverage for the sample. The authors added a sentence in the “methods” section.
“The collection of information on vaccination in our study was not foreseen in the methodology developed in July 2020 due to a lack of information on the availability of a vaccine for staff in 2021.”
Moreover, we were able to collect information on COVID-19 vaccination coverage of nursing home staff in the Loire department.
“The data from Santé Publique France, on March 16,2021, showed that 20.8% of professionals working in retirement homes in the Loire had received a first dose of vaccine. It is estimated that 74.6% of the residents of retirement homes located in the Loire have received a first dose of the COVID-19 vaccine.”
Admission
1) In the initial stage of the introduction, the authors provide statistical data without any information about the source - please complete the bibliography
- Response to reviewer 2: The bibliography has been completed to indicate the source of data.
(1) SPF. COVID-19 : point épidémiologique du 31 décembre 2020 https://www.santepubliquefrance.fr/maladies-et-traumatismes/maladies-et-infections-respiratoires/infection-a-coronavirus/documents/bulletin-national/covid-19-point-epidemiologique-du-31-decembre-2020 (accessed 2022 -03 -21).
(2) SPF. COVID-19 : point épidémiologique du 11 juin 2020 /maladies-et-traumatismes/maladies-et-infections-respiratoires/infection-a-coronavirus/documents/bulletin-national/covid-19-point-epidemiologique-du-11-juin-2020 (accessed 2020 -06 -18).
2) At the end of the introduction, the research hypotheses on which the research was created with the source given should be presented
- Response to reviewer 2: the research hypotheses on which the research was created with the source given have been added by the authors in the introduction
“The French High Authority of Health published recommendations, validated on May 7, 2020, aimed at preventing and identifying occupational suffering in the health sector, with guidance in the context of the COVID 19 crisis [16]. Health professionals are in first line in the management of the epidemic in healthcare facilities. Considering that humanity is undergoing the most severe pandemic since the Spanish flu, the current COVID-19 pandemic is very likely to promote PTSD, which commonly occurs during major disasters [17,18]. The French National Academy of Medicine, on June 8, 2020, recommended that special long-term attention be paid to the mental health of caregivers involved in the management of COVID-19, who should have systematic medical examination and 3 years’ follow-up by preventive medicine doctors to identify "possible psychological symptoms that appeared after the acute phase of the health crisis"[15]. Currently, the medical follow-up of caregivers by the occupational health service is based on medical consultations [19]. Due to a lack of occupational physicians in healthcare institutions, systematic consultation would be very difficult to implement in the short term [20] “
(15) Suivi Des Soignants Impliqués Dans La Prise En Charge de La COVID-19. Communiqué de l’Académie nationale de Médecine. June 8, 2020.
(16) HAS • Réponse Rapide Dans Le Cadre Du COVID-19 - Souffrance Des Professionnels Du Monde de La Santé : Prévenir, Repérer, Orienter • Mai 2020.
(17) Cai, H.; Tu, B.; Ma, J.; Chen, L.; Fu, L.; Jiang, Y.; Zhuang, Q. Psychological Impact and Coping Strategies of Frontline Medical Staff in Hunan Between January and March 2020 During the Outbreak of Coronavirus Disease 2019 (COVID‑19) in Hubei, China. Med Sci Monit 2020, 26. https://doi.org/10.12659/MSM.924171.
(18) Gold, J. A. Covid-19: Adverse Mental Health Outcomes for Healthcare Workers. BMJ 2020, 369, m1815. https://doi.org/10.1136/bmj.m1815.
(19) Le suivi de l’état de santé des salariés https://travail-emploi.gouv.fr/sante-au-travail/suivi-de-la-sante-au-travail-10727/article/le-suivi-de-l-etat-de-sante-des-salaries (accessed 2022 -03 -19).
(20) Marichalar, P. «â€¯La Médecine Du Travail sans Les Médecins ? Une Action Patronale de Longue Haleine (1971-2010) ». Politix 2010, 3 (91), 27–52.
Results
1) In my opinion, to be removed completely section 3.2. this will also translate to a change of title and targets with the deletion of information on qualitative measurements.
- Response to reviewer 2: Following reviewers’ comments, the “results” have been modified in the text and in the tables:
- The results about qualitative results have been put at the start as pilot work.
- The previous tables 2, 3 and 4 have been deleted. New tables have been created to show the results after univariate analysis (table 2) and after multivariate analysis (table 3). Parts of this text have been deleted.
2) Why did the authors choose PR instead of OR?
- Response to reviewer 2:
The authors have added a sentence in “methods” to justify their choice
“As the prevalence of the events (anxiety, depression, or PTSD) were high, odds ratios would not provide a good estimate of relative risk rather, the log-linked binomial model were applied, using the PROC GENMOD procedure in the SAS statistical package (version 9.4) with the DIST=BINOMIAL and LINK=LOG options.”
3) Figure 1 - very poor quality, correction needed.
- Response to reviewer 2: The authors have modified the figure 1 in order to improve it.
Discussion
1) The discussion does not include a complete discussion of the influence of sociodemographic variables on the mental state, including age and gender, with a discussion of the results with world data.
- Response to reviewer 2:
The authors have added in the results information about relationship between mental disorder and sociodemographic variables:
“In contrast, anxiety symptoms were not significantly associated with gender (p=0.8) and age (p=0.4) and family situation (P=0.3)”
“In contrast, depressive symptoms were not significantly associated with gender (p=0.8) and age (p=0.8) and family situation (P=0.9)”
“In contrast, depressive signs of PTSD were not significantly associated with gender (p=0.7) and age (p=0.8) and family situation (P=0.7)”
Moreover, the authors have added in the discussion sentences to discuss the influence of sociodemographic variables on the mental state.
“In our study, anxiety symptoms, depressive symptoms were not significantly associated with gender which is inconsistent with the data in the literature [49–51]. Moreover, anxiety symptoms and depressive symptoms, were not significantly associated with age whereas, differences by age were reported, with younger Health Care Workers experiencing higher levels of anxiety and depressive symptoms compared to older age groups in the literature [52–54].“
(49) Liu, S.; Yang, L.; Zhang, C.; Xu, Y.; Cai, L.; Ma, S.; Wang, Y.; Cai, Z.; Du, H.; Li, R.; Kang, L.; Zheng, H.; Liu, Z.; Zhang, B. Gender Differences in Mental Health Problems of Healthcare Workers during the Coronavirus Disease 2019 Outbreak. Journal of Psychiatric Research 2021, 137, 393–400. https://doi.org/10.1016/j.jpsychires.2021.03.014.
(50) De Sio, S.; Buomprisco, G.; La Torre, G.; Lapteva, E.; Perri, R.; Greco, E.; Mucci, N.; Cedrone, F. The Impact of COVID-19 on Doctors’ Well-Being: Results of a Web Survey during the Lockdown in Italy. Eur Rev Med Pharmacol Sci 2020, 24 (14), 7869–7879. https://doi.org/10.26355/eurrev_202007_22292.
(51) Lai, J.; Ma, S.; Wang, Y.; Cai, Z.; Hu, J.; Wei, N.; Wu, J.; Du, H.; Chen, T.; Li, R.; Tan, H.; Kang, L.; Yao, L.; Huang, M.; Wang, H.; Wang, G.; Liu, Z.; Hu, S. Factors Associated With Mental Health Outcomes Among Health Care Workers Exposed to Coronavirus Disease 2019. JAMA Network Open 2020, 3 (3), e203976. https://doi.org/10.1001/jamanetworkopen.2020.3976.
(52) Rossi, R.; Socci, V.; Pacitti, F.; Di Lorenzo, G.; Di Marco, A.; Siracusano, A.; Rossi, A. Mental Health Outcomes Among Frontline and Second-Line Health Care Workers During the Coronavirus Disease 2019 (COVID-19) Pandemic in Italy. JAMA Network Open 2020, 3 (5), e2010185. https://doi.org/10.1001/jamanetworkopen.2020.10185.
(53) Huang J. Mental health survey of 230 medical staff in a tertiary infectious disease hospital for COVID-19. Chinese Journal of Industrial Hygiene and Occupational Diseases 2020, E001–E001.
(54) Salman, M.; Raza, M. H.; Ul Mustafa, Z.; Khan, T. M.; Asif, N.; Tahir, H.; Shehzadi, N.; Hussain, K. The Psychological Effects of COVID-19 on Frontline Healthcare Workers and How They Are Coping: A Web-Based, Cross-Sectional Study from Pakista. 2020. https://doi.org/10.1101/2020.06.03.20119867.
2) I reiterate my question about the vaccination profile - if so, discuss it, if not add to the test limits as research shows that vaccination can have an impact on mental health.
- Response to reviewer 2:
The authors added a paragraph in the study's limitations.
“Vaccination against COVID-19 may have had an effect on the prevalence of anxiety, depressive and post-traumatic stress disorder in nursing home staff involved in the COVID-19 health crisis. According to Haddaden et al., a COVID-19 vaccination conferred improvement in Healthcare workers comfort and anxiety in caring for patients with COVID-19 and other illnesses likely explained by the high efficacy of vaccination in preventing symptomatic infection and critical illness [66]. The COVID-19 vaccination coverage of the sample was not known. However, the data from Santé Publique France, on March 16,2021, showed that 20.8% of professionals working in retirement homes in the Loire had received a first dose of vaccine. It is estimated that 74.6% of the residents of retirement homes located in the Loire have received a first dose of the COVID-19 vaccine [67].”
3) The authors present the results from other countries, periods of the pandemic and before - was the impact of direct combat with the pandemic on health assessed? If so please discuss if not re-restrictions.
- Response to reviewer 2:
The authors added the following sentences:
“Studies that examined mental health outcomes in healthcare workers before and during the COVID-19 outbreak found a significant increase in reported anxiety symptoms in the outbreak period compared to the non-outbreak period [34,35].”
“Previous studies highlighted an important impact of the COVID-19 pandemic on the mental health of healthcare workers[41]”
(34) Lu, W.; Wang, H.; Lin, Y.; Li, L. Psychological Status of Medical Workforce during the COVID-19 Pandemic: A Cross-Sectional Study. Psychiatry Res 2020, 288, 112936. https://doi.org/10.1016/j.psychres.2020.112936.
(35) Li, W.; Frank, E.; Zhao, Z.; Chen, L.; Wang, Z.; Burmeister, M.; Sen, S. Mental Health of Young Physicians in China During the Novel Coronavirus Disease 2019 Outbreak. JAMA Netw Open 2020, e2010705–e2010705.
(41) SPF. La souffrance psychique en lien avec le travail chez les salariés actifs en France entre 2007 et 2012, à partir du programme MCP https://www.santepubliquefrance.fr/maladies-et-traumatismes/maladies-liees-au-travail/souffrance-psychique-et-epuisement-professionnel/la-souffrance-psychique-en-lien-avec-le-travail-chez-les-salaries-actifs-en-france-entre-2007-et-2012-a-partir-du-programme-mcp (accessed 2021 -12 -16).
Reviewer 3 Report
This study is potentially interesting and socially useful, but it does need considerable revision in term of presentation and analysis.
Abstract:
Line 10 – how is this an ‘observational’ study?
Line 19 – here and in many places subsequently, we are told about ‘the questionnaire’. But this is never well defined. It appears to incorporate the HAD and PCL-5 and demographics, and perceived stress …but (later in Method) we need a clearer enumerated description.
Introduction:
Lines 55 and 62 – make clear these bodies are in France (if they are!).
The Aims are clear – but I would suggest putting the qualitative study first – this gives us some confidence in ‘the questionnaire’ – then give the quantitative findings.
Method:
Line 87 – give is some idea of where the Saint-Etienne areas of France is (and maybe some background e.g. urban/rural, income level etc).
Later – especially lines 130-135 – we need more detail, presented systematically as stated earlier.
Lines 136-144 – as above I would put this qualitative study separately and earlier – almost as pilot work.
Results:
There are a lot of Tables. However I did not always find a clear link from factors (e.g. history of anxiety disorder) to how this is defined (should be in Method). The Method should clearly list and define the factors in the Tables.
The material in the text repeats percentages etc in the tables. This is unnecessary. Either shorten the text, or delete the tables.
Line 182 – univariate analyses are used for many factors. This is unfortunate. Multivariate analyses should be carried out. If not, Bonferroni corrections should be made for the number of statistical tests made.
Line 238 - this qualitative study best comes earlier as pilot work or a first study.
Author Response
We thank the three reviewers for their suggestions of modifications which contribute to improve the article
Reviewer 3
This study is potentially interesting and socially useful, but it does need considerable revision in term of presentation and analysis.
Abstract:
Line 10 – how is this an ‘observational’ study?
- Response to reviewer 3:
This is an epidemiological survey. The authors have changed “observational” by “epidemiological” in the introduction.
“A multicenter epidemiological study was conducted in 12 nursing homes in France with 1117 nursing home staff eligible”
Line 19 – here and in many places subsequently, we are told about ‘the questionnaire’. But this is never well defined. It appears to incorporate the HAD and PCL-5 and demographics, and perceived stress …but (later in Method) we need a clearer enumerated description.
- Response to reviewer 3:
The authors have modified the abstract to underline the use of validated scales in the questionnaire to assess mental disorder. However, more details have been added in “methods” because of the abstract should be a total of about 200 words maximum.
“Socio-demographic, occupational and medical data were collected by anonymous self-reported questionnaire using validated scales to assess anxiety/depressive symptoms (HAD scale) and post-traumatic stress disorder (PCL-5).”
Introduction:
Lines 55 and 62 – make clear these bodies are in France (if they are!).
- Response to reviewer 3:
The authors have specified the French origin for High Authority of Health and National Academy of Medicine.
“The French High Authority of Health published recommendations, validated on May 7, 2020, aimed at preventing and identifying occupational suffering in the health sector, with guidance in the context of the COVID 19 crisis [16]. Health professionals are in first line in the management of the epidemic in healthcare facilities. Considering that humanity is undergoing the most severe pandemic since the Spanish flu, the current COVID-19 pandemic is very likely to promote PTSD, which commonly occurs during major disasters [17,18]. The French National Academy of Medicine, on June 8, 2020, recommended that special long-term attention be paid to the mental health of caregivers involved in the management of COVID-19, who should have systematic medical examination and 3 years’ follow-up by preventive medicine doctors to identify "possible psychological symptoms that appeared after the acute phase of the health crisis"[15].”
(15) Suivi Des Soignants Impliqués Dans La Prise En Charge de La COVID-19. Communiqué de l’Académie nationale de Médecine. June 8, 2020.
(16) HAS • Réponse Rapide Dans Le Cadre Du COVID-19 - Souffrance Des Professionnels Du Monde de La Santé : Prévenir, Repérer, Orienter • Mai 2020.
The Aims are clear – but I would suggest putting the qualitative study first – this gives us some confidence in ‘the questionnaire’ – then give the quantitative findings.
- Response to reviewer 3: The authors have modified the text to put the qualitative study first.
“The purpose of this study was to evaluate qualitatively and quantitatively the use of a self-reported questionnaire as a tool for screening for mental disorder in nursing home staff.
Study objectives were:
- to evaluate the acceptance and expectations regarding screening by self-reported questionnaire;
- to determine the prevalence of anxiety, depressive and post-traumatic stress disorder in nursing home staff involved in the COVID-19 health crisis;
- and to identify occupational and medical factors associated with anxiety, depression and post-traumatic stress disorder symptoms, in order to adapt prevention actions.”
Method:
Line 87 – give is some idea of where the Saint-Etienne areas of France is (and maybe some background e.g. urban/rural, income level etc).
- Response to reviewer 3: The authors have added the following sentence
“Saint Etienne is a city of 173,089 inhabitants located in the Loire department in France. 72% of the population has a level of education equivalent or lower than the baccalaureate. In 2019, the median income was 18410 euros.”
Later – especially lines 130-135 – we need more detail, presented systematically as stated earlier.
- Response to reviewer 3:
The authors have added more detail to present variables:
“In a context of COVID-19 health crisis, socio-occupational and medical factors associated with these mental pathologies were sought on the basis of questions included in the questionnaire:
- Socio-occupational factors: age, gender, family situation, job, working hours, seniority
- experience of the COVID crisis:
- Difficulties in reconciling family and working life
- Feeling of inadequate protection against COVID-19 infectious risk
- Feeling of concern about the risk of transmission of COVID-19 for residents, for relatives, for their own health, for colleagues and superiors
- Feeling of difficulties in accompanying residents with COVID-19 or their families
- due to lack of equipment
- due to lack of communication within the facility
- due to a feeling of dehumanization of care
- due to lack of time
- due to a feeling of powerlessness
- Feeling of difficulties in accompanying residents with COVID-19 or their families
- Confrontation with the deaths of COVID-19 residents and emotional affects
- Feeling of a traumatic work-related event during the health crisis
- Medical factors:
- History of anxiety disorder
- History of depressive syndrome
- psychotropic treatment
- work stoppage in the last 12 months
- Duration of the work stoppage
- Feeling of a traumatic work-related event during the health crisis”
Lines 136-144 – as above I would put this qualitative study separately and earlier – almost as pilot work.
- Response to reviewer 3:
The authors have changed the description of the qualitative study in the beginning of “Methods” as pilot work.
Results:
There are a lot of Tables. However, I did not always find a clear link from factors (e.g. history of anxiety disorder) to how this is defined (should be in Method). The Method should clearly list and define the factors in the Tables.
The material in the text repeats percentages etc in the tables. This is unnecessary. Either shorten the text, or delete the tables.
- Response to reviewer 3:
The authors have modified the tables and text to clarify the results: the previous tables 2, 3 and 4 have been deleted. New tables have been created to show the results after univariate analysis (table 2) and after multivariate analysis (table 3). Parts of this text have been deleted.
Line 182 – univariate analyses are used for many factors. This is unfortunate. Multivariate analyses should be carried out. If not, Bonferroni corrections should be made for the number of statistical tests made.
- Response to reviewer 3:
The authors have added a multivariate analysis. Multivariable analyses were performed for each outcome (anxiety, depression, PTSD), variables with a p-value<=0 .05 in the univariate step were included in a multivariate model by a descending procedure and variables with p-values < 0.05 remained in the model. The results have been showed in the table 3.
Line 238 - this qualitative study best comes earlier as pilot work or a first study
Response to reviewer 3: The authors have put the results of qualitative study at the start.
Round 2
Reviewer 1 Report
In this modified version, the authors basically answered the questions I raised in the first version, but the modified format is too bad. From the current version, in my opinion, the author still lists the data analysis results line by line, lacking a general summary.
Author Response
The authors would like to thank the reviewer for his comments, which helped to improve the article.
They have added sentences to summarize the results:
"Also according to this qualitative assessment, the questionnaire was well accepted by the participants and met their wishes for prevention action."
"Although the results of the univariate analysis show an association between difficulties in reconciling family /working life imbalance and anxiety symptoms, depressive symptoms or PTSD, the association is not maintained after multivariate analysis. On the other hand, medical factors (such as history of anxiety or depressive symptoms), and occupational factors (such as high occupational stress) remained associated with these pathologies after multivariate analysis in nursing home staff involved in the COVID-19 health crisis."
Reviewer 2 Report
The authors addressed all suggestions. The article meets the publication criteria.
Author Response
The authors would like to thank the reviewer for his or her comments, which helped to improve the article.
Reviewer 3 Report
The ms. has been greatly improved in revision. I find it now acceptable for publication.
Author Response

(The authors gave the same response as above.)
